# Neutralization mechanism of a highly potent antibody against Zika virus

Shuijun Zhang[1,2,*], Victor A. Kostyuchenko[1,2,*], Thiam-Seng Ng[1,2,*], Xin-Ni Lim[1,2], Justin S.G. Ooi[1,2], Sebastian Lambert[1,2], Ter Yong Tan[1,2], Douglas G. Widman[3], Jian Shi[2,4], Ralph S. Baric[3] & Shee-Mei Lok[1,2]

The rapid spread of Zika virus (ZIKV), which causes microcephaly and Guillain-Barré syndrome, signals an urgency to identify therapeutics. Recent efforts to rescreen dengue virus human antibodies for ZIKV cross-neutralization activity showed antibody C10 as one of the most potent. To investigate the ability of the antibody to block fusion, we determined the cryoEM structures of the C10-ZIKV complex at pH levels mimicking the extracellular (pH8.0), early (pH6.5) and late endosomal (pH5.0) environments. The 4.0 Å resolution pH8.0 complex structure shows that the antibody binds to E proteins residues at the intra-dimer interface, and the virus quaternary structure-dependent inter-dimer and inter-raft interfaces. At pH6.5, antibody C10 locks all virus surface E proteins, and at pH5.0, it locks the E protein raft structure, suggesting that it prevents the structural rearrangement of the E proteins during the fusion event—a vital step for infection. This suggests antibody C10 could be a good therapeutic candidate.

[1] Program in Emerging Infectious Diseases, Duke–National University of Singapore Medical School, Singapore 169857, Singapore. [2] Centre for BioImaging Sciences, National University of Singapore, Singapore 117557, Singapore. [3] Department of Epidemiology, University of North Carolina at Chapel Hill, Chapel Hill, North Carolina 27599-7435, USA. [4] CryoEM unit, Department of Biological Sciences, National University of Singapore, Singapore 117557, Singapore. * These authors contributed equally to this work. Correspondence and requests for materials should be addressed to R.S.B. (email: rbaric@email.unc.edu) or to S.-M.L. (email: sheemei.lok@duke-nus.edu.sg).

Zika virus[1] (ZIKV) is a member of the flavivirus genus that includes dengue virus (DENV) and West Nile virus (WNV). ZIKV cryoEM structures[2,3] show its surface proteins (envelope (E) and membrane (M) proteins) are organized similar to DENV[4] except with a tighter packing, making the virus more thermally stable[2].

The virus surface consists of 180 copies of E protein[2] arranged in icosahedral symmetry with 60 asymmetric units. In each asymmetric unit, there are three individual E proteins – molecules A, B and C. The E proteins exist as dimers; three dimers lie parallel to each other forming a raft containing two asymmetric units. There are in total 30 rafts arranged in a herringbone pattern on the virus surface.

An E protein contains three domains—DI, DII and DIII[5]. It is known for other flaviviruses that DIII contains the receptor-binding site and plays an important role in fusion of the virus with the endosomal membrane during cell entry[6,7]. The tip of DII contains a fusion loop that interacts with the endosomal membrane. DI is the central domain linking DII and DIII together. The DI-DII hinge is highly flexible allowing DII to expose its fusion loop during the fusion event. The DI-DIII hinge was thought to be more rigid but it was observed to change in conformation in the post-fusion E protein trimeric structure[6,7]. The fusion event is hypothesized to occur in this sequence: (1) virus E protein binds to cell receptors, (2) it is endocytosed, (3) the low pH environment of the endosome causes the E proteins to flip up exposing their fusion loops, allowing them to interact with the endosomal membrane, (4) the E proteins rearrange to trimeric structures, (5) the DIIIs of the E protein trimers change in conformation twisting the trimers leading to the fusion of viral membrane with the endosomal membrane, before the release of the viral genome into cell cytosol.

The recent explosion of the number of ZIKV cases, together with the association of ZIKV with the development of microcephaly in fetuses[8] and Guillian-Barré syndrome in adults[9], ignite a pressing need for the development of therapeutics. Currently there are no published human monoclonal antibodies (HMAb) generated against ZIKV. To hasten the process of therapeutics development, DENV HMAbs were rescreened[10–12] for those that cross-neutralize ZIKV. One group of antibodies has recently been shown to be highly neutralizing to ZIKV—the envelope dimer epitope binding antibodies[10,11]. Of these HMAbs, C10 is one of the most potent plaque reduction neutralisation test ($PRNT_{50} = 0.024\,\mu g\,ml^{-1}$), as demonstrated recently in ZIKV infected cell culture[11,13] and mouse model[13]. In addition, it can prevent antibody dependent enhancement (ADE) of ZIKV infection in myeloid cells induced by dengue human sera[10]. In this ADE model, the myeloid cells are mostly resistant to direct ZIKV infection, suggesting that its specific receptor is lacking. When sub-neutralizing concentrations of dengue human serum was added to ZIKV, cell infection was enhanced. This is because antibodies, which are attached to ZIKV, bind to the Fc receptor on myeloid cells thus bypassing the need for ZIKV to directly interact with its specific receptor. When HMAb C10 is added to this mixture, it neutralizes the ADE effect. Since HMAb C10 is also an antibody that would likely facilitate attachment to Fc receptor on myeloid cells, it likely neutralizes the virus at a post-attachment step of infection. We investigated the ability of Fab C10 to prevent virus surface protein rearrangement during fusion. We observed Fab C10 is able to lock the entire virus surface at pH6.5, and at pH5.0, the E protein raft thereby preventing structural rearrangement necessary for fusion.

## Results

**Effect of Fab C10 on ZIKV particles at different pHs**. We solved the cryoEM structures of Fab C10 complexed with ZIKV at pH8.0, pH6.5 and pH5.0 mimicking the extracellular, early and late endosomal conditions, respectively, and compared them to the cryoEM maps of the uncomplexed ZIKV controls at pH8.0 (ref. 2), pH6.5 (Supplementary Fig. 1b) and the two-dimensional (2D)-class average of pH5.0 particles (Fig. 1).

Micrographs of the uncomplexed ZIKV control at pH8.0 sample show mostly smooth surfaced spherical particles (Fig. 1). In the pH6.5 control sample (Fig. 1), some virus particles aggregated, others become deformed, but there are also spherical particles present. 2D class average of the pH6.5 spherical particles (Fig. 1 inset), as well as its low resolution cryoEM map

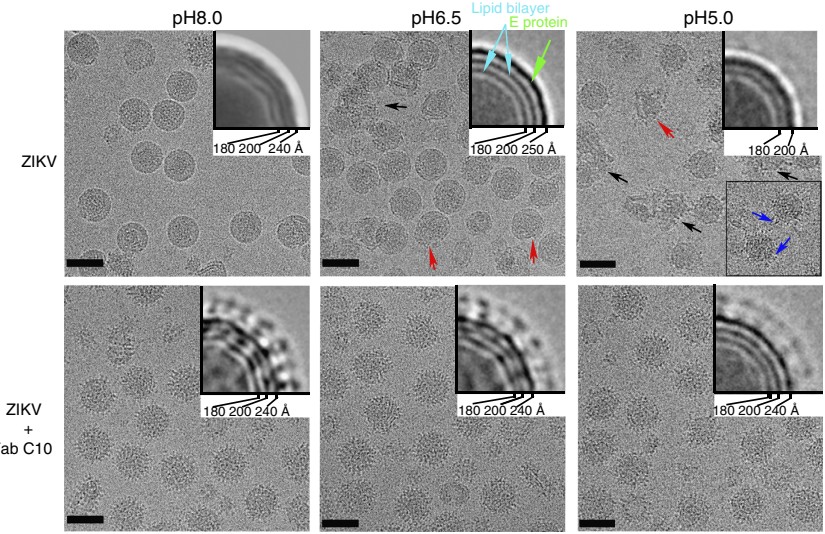

**Figure 1 | CryoEM micrographs of the uncomplexed ZIKV control and the Fab C10-ZIKV complex samples at various pH levels.** The deformed particles and aggregates are indicated with red and black arrows, respectively. The right upper corner inset shows a quarter of a 2D class average of the round particles. The E protein layer is indicated with a green arrow, the outer and inner leaflets of the bilayer lipid membrane with cyan arrows. In the pH5.0 uncomplexed ZIKV control, the E protein layer is missing in the 2D class average. Bottom right inset in the pH5.0 uncomplexed ZIKV control is a median filtered (5 × 5 pixel) image that showed particles with hair-like protrusions (blue arrow), which are likely the E proteins flopping on the virus surface. Scale bar is 500 Å.

(Supplementary Fig. 1a and b), show the outer E protein layer has moved to a slightly larger radius compared to the pH8.0 control virus. This suggests that the E protein layer has loosened. Micrographs of the pH5.0 control sample (Fig. 1) show aggregation of some particles, while others appear to be smaller in diameter with hair-like densities protruding from the virus surface. The 2D class average of these small particles (Fig. 1 inset) showed the absence of the E protein compact layer, which was present in the pH8.0 and pH6.5 control samples. This suggests the E proteins are likely 'flopping' on the virus surface. The ZIKV controls demonstrate some of the structural transformation stages of the virus particles during fusion, from the compact structure at pH8.0 to a slightly expanded structure at pH6.5 and finally to the E proteins loosening and extending out from the virus lipid membrane at pH5.0.

Micrographs of the ZIKV-C10 complexes at all pH conditions show spiky looking particles, due to the Fab molecules bound to virus surface (Fig. 1). The 2D class average of the pH6.5 complex particles shows the E protein layer to remain at the same radius as the pH8.0 control (Fig. 1 inset), unlike its pH6.5 ZIKV control. The 2D class average of the pH5.0 complex particles in contrast to its pH5.0 control shows the E protein layer is still present (Fig. 1 inset).

**CryoEM structures of ZIKV-C10 at different pHs.** The cryoEM structures of ZIKV-C10 complex at pH8.0, pH6.5 and pH5.0 are determined to 4.0, 4.4 and 12 Å resolution, respectively (Fig. 2, Supplementary Fig. 2). In each of these structures, there are 180 copies of Fab C10 bound to the virus surface. The pH8.0 and pH6.5 complex structures are very similar to each other (Supplementary Fig. 3a) and their cryoEM maps correlate to 4.5 Å resolution at FSC 0.143 cutoff (Supplementary Fig. 3b). Therefore, only the higher resolution pH8.0 complex structure will be described. A comparison of the E proteins of the pH8.0 complex structure with that of the previously solved uncomplexed ZIKV[2] shows that they are largely the same (Supplementary Fig. 4a). Only molecule A of the E protein in the asymmetric unit on the pH8.0 complex structure shows clear densities for the '150 glycan loop' (Supplementary Fig. 4b). The glycan loop changed in conformation when compared to the uncomplexed virus (Supplementary Fig. 4a, inset), likely due to its interaction with the Fab molecule (Supplementary Table 1). This glycan loop on ZIKV is five residues longer than in DENV. The previously solved crystal structure of DENV-C10 (ref. 14) did not show densities corresponding to the glycan loop; therefore, it is not known if this region interacts directly with the Fab. However, mutational studies[15] indicate that the residue 153 glycosylation site on DENV is not important for HMAb C10 binding.

In the 4.0 Å resolution pH8.0 complex cryoEM map (Fig. 2a), the likely interacting residues that form the epitope were identified, by using a cutoff of 5 Å distance[16] (hydrogen bonds/electrostatic interaction: 4 Å and hydrophobic interactions: 5 Å) between side chains of the Fab and E proteins (Fig. 3, Supplementary Fig. 5d, Supplementary Table 1). We also presented the epitope identified with a cutoff of 8 Å distance[17] between the Cα chains of the Fab and E proteins (Supplementary Fig. 5c). Each end of the E protein dimer has a Fab molecule attached (Fig. 3a). The Fabs bind across the E proteins at the intra-dimer interface (Fig. 3a, Supplementary Fig. 5b). The Fab bound near the five-fold vertex end of the A-C′ dimer also likely interacts with residues from the adjacent E protein at the inter-raft interface, whereas the Fab molecule at the other end is also involved in inter-dimer interactions within the raft (Fig. 3a, Supplementary

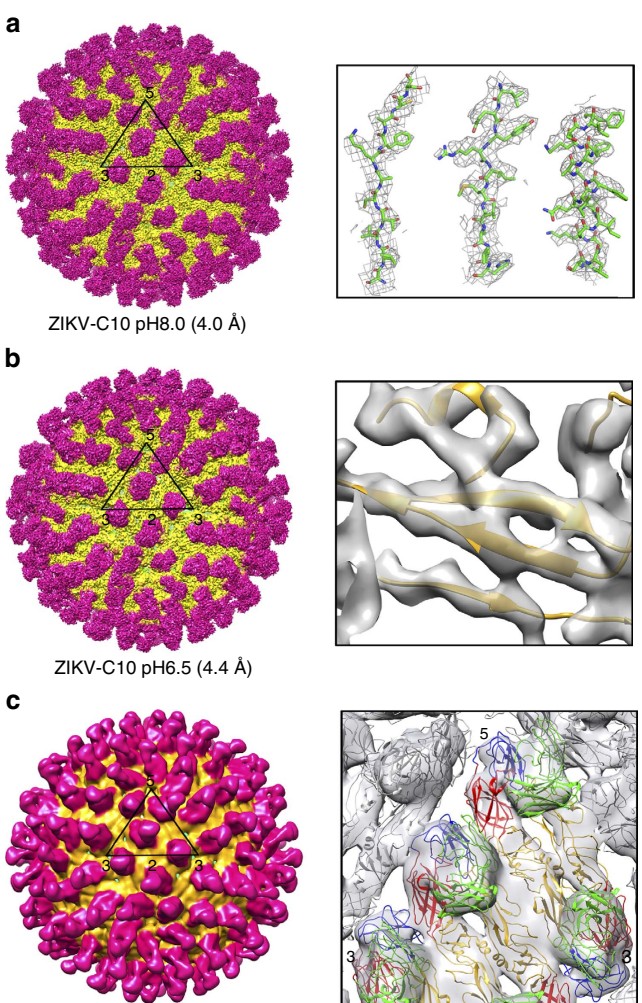

**Figure 2 | CryoEM maps of the Fab C10-ZIKV complex.** Structures at (**a**) pH8.0, (**b**) pH6.5, (**c**) pH5.0, determined to 4.0, 4.4 and 12 Å resolution, respectively. Left panels show the surface of the cryoEM maps. Densities corresponding to the E protein layer and Fabs are coloured in yellow and magenta, respectively. Black triangle indicates an asymmetric unit and the 5-, 3-, 2-fold vertices are labelled. Right panels show zoom-in views of the fitted molecules into the density maps. (**a**, right panel) The 4.0 Å resolution pH8.0 complex show well-resolved bulky side chain densities (grey mesh). The Cα backbone, the nitrogen and oxygen atoms are coloured in green, blue and red, respectively. (**b**, right panel) The 4.4 Å resolution pH6.5 complex map showed density (grey transparent surface) separation between the β strands. DII of E protein is coloured in yellow. (**c**, right panel) Densities of the 12 Å resolution pH5.0 complex showed clear borders and shapes corresponding to the Fab C10-E protein dimeric structures. The variable region of the Fab molecule, DI, DII and DIII of the E protein are coloured in green, red, yellow and blue, respectively.

Fig. 5b). The epitopes recognized by the Fab molecules that bind to B-B′ dimer also span across the inter-dimer E protein interfaces. The ability of Fab C10 to bind E proteins at the intra-dimer interface together with the virus quaternary structure-dependent sites—the inter-dimer and inter-raft interfaces, suggests that the entire E protein layer is locked. This is consistent with the cryoEM structure (Supplementary Fig. 3a) and the 2D class average (Fig. 1 inset) of the ZIKV-C10 complex at pH6.5 showing the E protein layer remains at a similar radius as the uncomplexed ZIKV pH8.0 control, unlike its pH6.5 control.

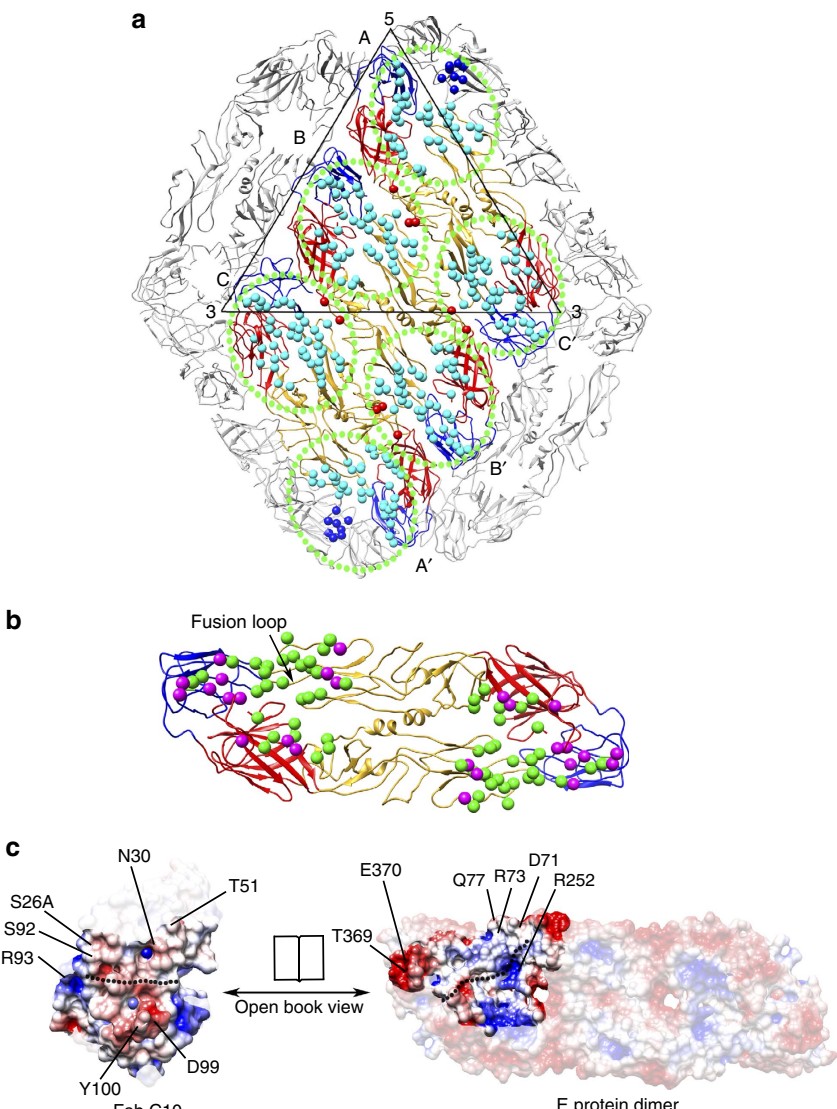

**Figure 3 | The C10 epitopes on the pH8.0 ZIKV-C10 complex structure. (a)** The C10 epitopes (circled by green dots) in an E protein raft identified by using a distance cutoff of 5 Å between the side chains of Fab and the E protein. The Fab molecules bind to both ends of each E protein dimer. The DI, DII and DIII of the E proteins in one raft are coloured in red, yellow and blue, respectively, those in neighbouring rafts are in grey. The three individual E proteins in an asymmetric unit are labelled as A, B and C molecules and those in the neighbouring asymmetric unit within the raft as A′, B′ and C′. The epitope residues within the intra-dimer interface are shown as light blue spheres, those at the inter-dimer and inter-raft interfaces as red and dark blue spheres, respectively. **(b)** The epitope within the intra-dimer interface on B-B′ dimer. The ZIKV c10 epitope residues that are conserved (similar charges or hydrophobicity) and non-conserved when compared to DENV are shown as green and magenta spheres, respectively. **(c)** Charge complementarity of the C10 intra-dimer epitope with the Fab paratope. Positive, negative and neutral charges are coloured in blue, red and white, respectively. Possible interacting residues are labelled.

## Discussion

The ZIKV-C10 complex intra-dimer epitope is located on the DII (at and around the fusion loop) on one E protein and on DIII and DI of the other E protein in the dimer (Fig. 3b). A plot of electrostatic charges of interacting residues on the E protein intra-dimer epitope and the Fab paratope showed complementary charges (Fig. 3c). A comparison with the previously published crystal structure of DENV recombinant E protein dimer complexed with Fab C10 (ref .14) (Supplementary Fig. 6b) shows their epitopes largely overlap. The conserved residues between ZIKV and DENV C10 intra-dimer epitope mainly cluster on DII near the fusion loop (Supplementary Fig. 6a and b). Although a comparison of the C10 intra-dimer epitope to the crystal structures of other EDE antibodies, Fab C8 and A11 complexed with ZIKV recombinant E protein[11]

(Supplementary Fig. 6c and d) shows overlapping epitopes, the C10 intra-dimer epitope spans a wider area covering larger parts of DI and DIII. Furthermore, our cryoEM structure also shows the interactions of Fab C10 with other virus quaternary structure-dependent epitope at the inter-dimer or inter-raft interfaces which are not observed in the crystal structures. These interactions result in the E proteins on virus surface being locked together and could be critical for its neutralization mechanism.

A comparison of the cryoEM maps of pH5.0 complex to the pH8.0 complex shows that the E protein layer has moved to a larger radius (Fig. 4a), whereas the radii of the lipid bilayer membranes are similar. The pH5.0 complex cryoEM map was interpreted by fitting separately the Fab:A-C′ dimer and Fab:B-B′ dimer structures from the pH8.0 complex structure into their respective densities

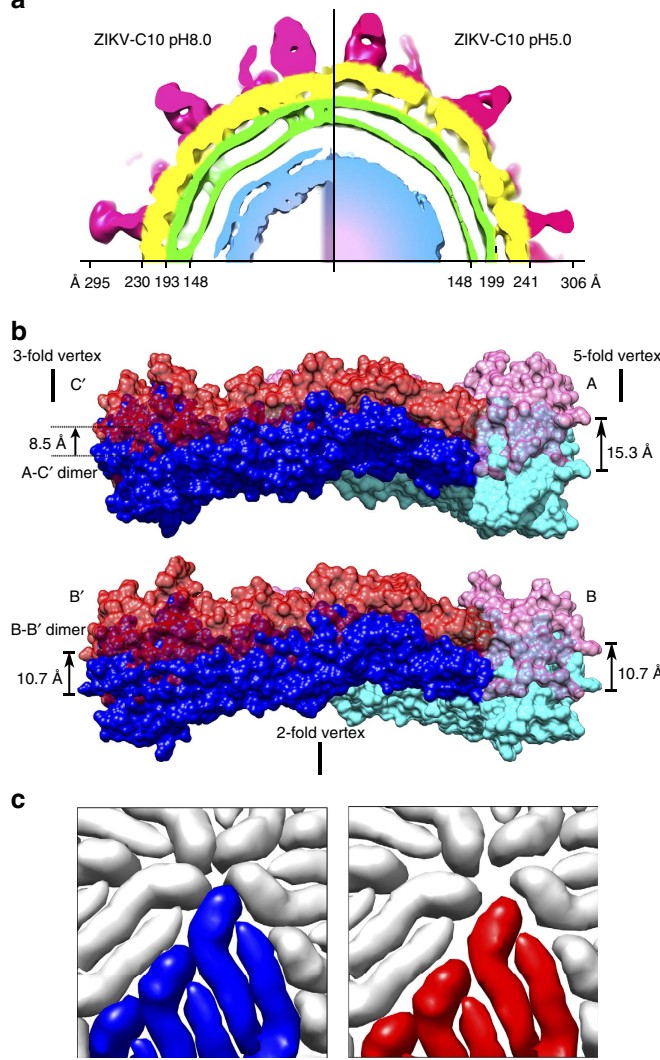

**Figure 4 | The radial movement of the E protein rafts in the ZIKV-C10 complex structure at pH5.0 compared to pH8.0.** (**a**) Comparison of a quarter of the cross-section of the pH8.0 and pH5.0 complex cryoEM maps. The pH8.0 complex cryoEM map is low-pass filtered to look similar to pH5.0 complex. The bilayer lipid membrane (green) of both maps is located at similar radii. The E protein layer (yellow) of the pH5.0 complex map, however, is at a larger radius. (**b**) Radial movement of A-C′ (top panel) and B-B′ (bottom panel) dimers in the pH5.0 complex structure compared to that at pH8.0. Side views of the dimers at pH5.0 (shades of red) and pH8.0 (shades of blue). The displacements of the ends of the dimer from pH8.0 to pH5.0 are indicated. Vertices are indicated. (**c**) The E protein inter-raft interactions of the pH5.0 complex structure are disrupted. One E protein raft of the pH8.0 and pH5.0 complex structures is coloured in blue and red, respectively, other surrounding rafts in grey. In the pH5.0 complex, the rafts are further apart from each other compared to the pH8.0 complex.

(Fig. 2c, right). Comparison of the E proteins in an asymmetric unit of the pH5.0 and pH8.0 complex by superimposing their molecules A shows a slight shift (3.5 Å) between the B′ molecule with respect to the A-C′ dimer (Supplementary Fig. 7). This motion has only slightly changed the distance of the interacting residues on the Fab with that on the E proteins at the inter-dimer interface (Supplementary Table 2) suggesting the Fab retains its binding capability across this interface at pH5.0 and thus the E proteins within the raft are locked by Fab C10.

Comparing the pH8.0 and the pH5.0 complex structures (Fig. 4b) shows that the maximum radial movement of the E protein outwards is at one end of the A-C′ dimer near the five-fold vertex (∼15 Å). This suggests the membrane associated stem regions of the E protein need not be fully extended (up to ∼65 Å in length) for this movement. In sharp contrast, a previous study[18] describing a very low resolution cryoEM map of a DIII-binding Fab E16:WNV complex at pH6.0 showed the E protein layer moved radially outwards by ∼60 Å, even though the E protein density was not interpretable. Our pH5.0 complex structure here shows a smaller radial expansion of the E protein layer and therefore may be an even earlier event of fusion process involving the dissociation of the E protein layer from the lipid membrane. Another low resolution cryoEM structure of antibody E104 complexed with DENV was shown to inhibit another stage of the fusion process, possibly the 'open trimeric E protein conformation'[19]. This is likely a step prior to the formation of the closed trimeric E protein structure[6].

Although the E protein raft structure stays mostly intact in the ZIKV-C10 pH5.0 complex structure, the inter-raft interactions are disrupted (Fig. 4c), even though the Fab C10 in the pH8.0 complex structure forms inter-raft interactions at two sites (Fig. 3a, Supplementary Fig. 5c). A calculation of the E protein electrostatic charges at pH8.0, 6.5 and 5.0 at the intra-dimer, inter-dimer and inter-raft interfaces (Supplementary Fig. 8) shows the residues becoming increasingly positively charged with decreasing pH. This suggests that at pH5.0, the E proteins may have the tendencies to repel each other, consistent with the 2D class average of the uncomplexed ZIKV particles at pH5.0 (Fig. 1 insets), showing that the compact E protein layer is disrupted. This raises a question of why the Fab inter-raft interaction in the pH5.0 complex structure is disrupted, whereas the inter-dimer interaction remains intact. We speculate that the two Fab molecules at the inter-raft interface (Fig. 3a) identified in the pH8.0 complex structure may not form strong enough contacts to resist the large surface area of electrostatic repelling force at this interface. On the other hand, at the inter-dimer interface, the two Fabs in this region could still hold the dimers together, as the surface area of repelling force is much smaller.

All antibodies can cause ADE at some concentrations. HMAb C10, similar to other potent antibodies, causes ADE at a much narrower range of concentrations compared to the other weakly neutralizing antibodies[10]. To increase the safety of HMAb C10 as a therapeutic antibody, its ability to cause ADE could be eliminated, by mutating its Fc region (LALA mutants) abolishing its interaction with the myeloid cells Fc receptors. Flavivirus such as DENV has been shown to be able to use different receptors to gain entry into different cell types[20,21]. ZIKV may also behave the same, as it has been shown to bind to DC-SIGN and also TAM receptors[22]. Since different parts of the E proteins interact with different receptors, it is unlikely that any single type of antibody could inhibit virus attachment to all cell types. In addition, ZIKV may also enter by ADE caused by pre-existing DENV antibodies in individuals, thus completely bypassing the need for virus to attach to a specific receptor for infection. However, regardless of how the virus gets into the cell, fusion is a vital step for productive infection. This emphasizes the potential of HMAb C10 as a therapeutic agent, since it can prevent structural rearrangements necessary for virus-endosomal membrane fusion.

## Methods

**Neutralization test of HMAb C10 to Zika virus.** The neutralization activity of the HMAb C10 on Zika virus strain H/PF/2013 was determined by PRNT. Two-fold serially diluted HMAb C10 starting at 0.5 µg ml$^{-1}$ were incubated with equal volumes of virus at 37 °C for 0.5 h. One hundred microlitres of each mixtures were

then layered on BHK-21 cells in a 24-well plate and incubated at 37 °C for 1.5 h. The infected cells were washed with phosphate-buffered saline, overlaid with carboxyl-methyl cellulose and incubated at 37 °C for 5 days. Cells were fixed and stained, and the plaques were counted. Percentage neutralization was determined from the comparison of the number of plaques in specific antibody dilutions to the control (without antibody). PRNT$_{50}$ is the concentration of the antibody that causes 50% reduction in plaque numbers.

**Virus sample preparation.** *Aedes albopictus* C6/36 cells (ATCC) were grown in RPMI 1640 media supplemented with 2% fetal bovine serum at 29 °C. At about 80% confluency, the cells were inoculated with ZIKV strain H/PF/2013 at a multiplicity of infection of 0.5 and incubated at 29 °C for 4 days. The virus-containing media was clarified by centrifugation at 12,000$g$ for 1 h. Virus was precipitated overnight from the supernatant using 8% (w/v) polyethylene glycol 8000 in NTE buffer (10 mM Tris-HCl pH8.0, 120 mM NaCl and 1 mM EDTA) and the suspension was centrifuged at 14,334$g$ for 1 h. The resulting pellet containing the virus was resuspended in NTE buffer and then purified through a 24% (w/v) sucrose cushion followed by a linear 10-30% w/v potassium tartrate gradient. The virus band, visualized by its light scattering ability, was extracted, buffer exchanged into NTE buffer and concentrated using a concentrator with 100-kDa molecular weight cut-off filter. All steps of the purification procedure were done at 4 °C. The concentrations and purity of the E protein were estimated with Coomassie blue-stained SDS-PAGE using different known concentrations of bovine serum albumin solution as standards.

**Monoclonal antibody C10 production.** Monoclonal antibody (mAb) C10 was synthesized in transfected human cells from cloned plasmids (Lake Pharma, Belmont, CA, USA). Briefly, previously published heavy and light chain variable region sequences[14] were cloned into plasmids containing the human IgG1 heavy and the human lambda 2 light chain constant regions. HEK293T cells (ATCC) were transiently co-transfected with both the heavy chain and light chain plasmids, and soluble antibody was collected and protein A purified. The antibody was resuspended in a buffer containing 200 mM HEPES, 100 mM NaCl, 50 mM NaOAc, pH7.0.

**Preparation of the C10 Fab fragments.** The Fab regions of C10 IgG were produced by papain digestion. Briefly, the whole IgG (8 mg ml$^{-1}$) was incubated overnight with immobilized papain (Thermo scientific) at 37 °C. After digestion, the Fab fragment was purified with anion exchange chromatography (resource Q, GE Healthcare) and gel filtration (Superdex 200 increase 10/300 GL, GE Healthcare) on an AKTA purifier system.

**CryoEM sample preparation.** The Fab C10 was mixed with ZIKV at a molar ratio of 1.5 Fab to every E protein. The mixture was incubated for 30 min at 37 °C followed by ∼1 h on ice, and then applied to a cryoEM grid (pre-cooled to 4 °C) for 10 s prior to adjusting the pH. The final pH of the virus was reached by addition of a volume ratio of 1.5 µl of 50 mM MES buffer at respective pH (pH5.0 or pH6.5) to every 1 µl of the virus-Fab mixture. The pH-adjusted samples were left on the grid for another 15 s. The grid was then blotted with filter paper and flash frozen in liquid ethane by using the Vitrobot Mark IV plunger (FEI, the Netherlands). The corresponding controls (ZIKV without Fab) for each pH were prepared similarly.

**Cryoelectron microscopy and image processing.** The images of the frozen ZIKV complexes were taken with the FEI Titan Krios electron microscope, equipped with 300 kV field emission gun, at nominal magnification of 47,000 for pH5.0 Fab C10 ZIKV complex, and 59,000 for pH6.5 and pH8.0 complex samples. A 4096 × 4096 FEI Falcon II direct electron detector was used to record the images.

Leginon[23] was used to carry out the data collection. Images for pH8.0 and pH6.5 complexes were collected in movie mode, with total exposure of 1.6 s and total dose 38 e$^{-}$ Å$^{-2}$ for pH8.0, total exposure of 1.05 s and total dose of 43 e$^{-}$ Å$^{-2}$ for pH6.5 complex. The pH5.0 complex was collected at single image mode, with the dose of 20 e$^{-}$ Å$^{-2}$. The frames from each 'movie' were aligned using MotionCorr[24] to produce full dose images used for particle selection and orientation search, and images from the first several frames amounting to the dose of about 18 e$^{-}$ Å$^{-2}$ to use in 3D reconstruction. The images were taken at underfocus in 0.5 ∼ 2.5 µm range. The astigmatic defocus parameters were estimated with Gctf[25] and accounted for in orientation search and 3D reconstruction procedures in MPSA[26] and Relion[27]. In total, 3,257, 2,540 and 2,865 micrographs were collected for Zika-C10 complex at pH8.0, pH6.5 and pH5.0, respectively. The virus-Fab particles were picked with automatic selection tool Gautomatch (from Dr K. Zhang, author of Gctf), run through 2D classification in Relion[27] to produce 2D class averages, broken and classes containing nonviral particles and broken particles were removed, 49,100, 45,867 and 23,810 individual particles in the Fab complex samples which were incubated at pH8.0, pH6.5 and pH5.0, respectively, were selected for further processing. The 3D reconstruction of the pH8.0 and pH6.5 complex structure was done with MPSA[26], whereas Relion was used for the pH5.0 complex structure. Uncomplexed ZIKV (EMDB ID EMD-8139) was used as the starting model. The gold standard protocol[27] for structure

refinement was used for all complexes. The 3D reconstruction procedure produced the complex structures with resolutions of 4.0, 4.4 and 12 Å, for pH8.0, pH6.5 and pH5.0, respectively—using the Fourier shell correlation cutoff of 0.143 for the pH8.0 and pH6.5 cryoEM maps and 0.5 for the pH5.0 cryoEM map (Extended Data Fig. 2).

**Protein structure building.** The pH8.0 and pH6.5 ZIKV-C10 structures were interpreted by fitting in the uncomplexed ZIKV (PDB ID 5IZ7) and Fab C10 (PDB ID 4UT9) first as rigid bodies in Chimera[28] and then finer adjustment were made by using the program Coot[29]. The 12 Å resolution cryoEM map of the ZIKV-C10 complex at pH5.0 was interpreted by first fitting in the entire E-protein raft complex with Fab molecules at pH8 structure by using the 'fit-in-map' function in Chimera. The individual Fabs complexed with A-C′ and also Fab complexed with B-B′ dimers within a raft were kept as separate rigid bodies groups and their fit into the density were independently optimized by using the 'fit-in-map' function in Chimera.

**Electrostatic potential calculations.** Electrostatic potentials of protein surfaces were calculated using Adaptive Poisson-Boltzmann Solver(APBS)[30] and PDB2PQR[31] packages. The structures of uncomplexed ZIKV (PDB ID 5IZ7) and the ZIKV complexed with Fab C10 were processed with the PDB2PQR web server (nbcr-222.ucsd.edu/pdb2pqr_2.0.0/) to prepare the PDB files for APBS. A PARSE force field was applied and PROPKA (v3.0) was used to assign pKa values. APBS was then used to calculate the electrostatic properties of the protein surface.

**Data availability.** The cryoEM maps and the atomic models of the ZIKV-C10 complex at pH8.0, 6.5 and 5.0 have been deposited in the Electron Microscopy Data Bank (EMD) and the Protein Data Bank (PDB) under the accession codes EMD-9575, EMD-9573, EMD-9574 and 5H37, 5H30, 5H32, respectively. The data that support the findings of this study are available from the corresponding authors on request.

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

## Acknowledgements

We thank the European Virus Archive for consenting to the use of the ZIKV H/PF/2013 strain and M.S. Diamond for sending the virus. The work is supported by Singapore Ministry of Education Tier 3 grant (MOE2012-T3-1-008), National Research Foundation Investigatorship award (NRF-NRFI2016-01) awarded to S-M.L and the Duke-NUS Signature Research Programme funded by the Ministry of Health, Singapore and NIH AID research grants AI100625 and AI 107731 awarded to R.S.B.

## Author contributions

X.-N.L. prepared the ZIKV particles. D.G.W. and S.Z. produced the HMAb C10 and the Fab fragment, respectively. X.-N.L. carried out the neutralization test, and together with T.-S.N. optimized the cryoEM freezing conditions. T.-S.N. and J.S. collected the cryoEM data. J.S.G.O., S.L., T.Y.T., S.Z. and V.A.K. processed cryoEM data. S.Z., V.A.K., J.S.G.O., X.-N.L., D.G.W., T.-S.N. prepared the material and methods section. R.S.B. and S-M.L. designed the experiments. S.-M.L., S.Z., J.S.G.O. and V.A.K. analysed the data and wrote the manuscript. S.L. helped edit the paper.

## Additional information

**Competing financial interests:** The authors declare no competing financial interests.

**Publisher's note**: 

