## [Peer Review File · Nature Communications]

Reviewers' comments:

Reviewer #1 (Remarks to the Author):

This manuscript describes cryo-EM analysis of Zika virus in complex with a neutralizing human monoclonal antibody, that was previously shown to neutralize dengue virus, at different pHs to gain insight into the mechanism of how this cross-reactive antibody neutralizes Zika virus infection in the context of membrane fusion. Authors first provide an analysis of cryo-EM images of the un-complexed native Zika virions at different pHs and show that lowering of pH from 8 to 6.5 radially extends the outer E protein layer, and at pH 5.5, this layer loosens further and extends out from the virion lipid layer. These observations are then compared with cryo-EM reconstructions of Zika virus in complex with the antibody at pH of 8, 6.5, and 5.5. These reconstructions are performed to a resolution of 4.0, 4.4 and 12 Å at these pH values, respectively. They clearly show that the antibody binds to the E protein layer at all these pHs. Comparative analysis of the complex structures at these pHs, and with that of cryo-EM image analysis of uncomplexed virions shows that this particular antibody constrains the movement of the E protein at pH 6.5 and further restricts structural rearrangement of the E protein subunits critical for the membrane fusion event.

The manuscript describes a set of novel and interesting structural results, and raises the possibility of using this kind of a cross-reactive antibody in designing neutralizing immunotherapeutic antibodies that are also effective in preventing the antibody-mediated enhancement (ADE effect). The manuscript is fairly well-written with adequate methodological details and appropriate discussion. The cryo-EM analysis is well done and results are convincing, and interpretations are reasonable.

My other major comments:

1. There are no structural details of the bound Fab presented in the main text. Authors MUST provide a summary of paratope-epitope interactions in the main text (which CDRs, residues etc) , and a figure showing how well the Fab density is resolved (either in the text or in the extended data). Extended data do show the paratope-epitope interactions, but a figure of close up view of the interacting regions would be useful. It is not clear why closer interactions are not listed, one would think at near ~4 Å resolution, with a reasonable atomic model, one would be able to 'see' hydrogen bond and other closer interactions.
2. Provide further cryo-EM imaging details – a) range of defocus values used during imaging and more details about the CTF corrections (important because the reconstructions at different

resolution are compared) and b) how the magnification was calibrated (important because the observations pertain to radial expansions).

3. Line 149: be more specific – which “another stage”

4. Line: 161: be more specific about which “repelling force at this interface”

5. Include a succinct more cogent discussion at the end that summarizes how the structural observations relate to fusion event sequence mentioned in the introduction.

6. Consider including comparative radial plots (with both uncomplexed and complexed structures) which would be more effective in illustrating the radial changes.

Reviewer #3 (Remarks to the Author):

The manuscript titled, “Neutralization mechanism of a highly potent antibody against Zika virus” by Shee-Mei Lok and colleagues describes the cryo-EM structures of the C10-ZIKV complex at different pH levels. While this study is not the first antibody-bound ZIKV structure solved, it represents a novel attempt to describe the antibody-bound structure in a biologically relevant way using varied pH levels to mimic the environment likely to be encountered by the virus during an infection cycle. The authors found that unbound ZIKV goes through structural changes as the pH decreases, showing deformed and aggregated viruses and losing structural stability of the E surface protein layer. However, when the C10 Fab is pre-bound to ZIKV, the E protein layer becomes stable, even at low pH. The authors estimated Fab-binding epitopes, and hypothesize that the locations are appropriate for stabilizing inter-dimer, intra-dimer and inter-raft protein interfaces. The authors point to this stabilization as a likely mechanism for virus neutralization by this Fab.

This study is novel and timely, as the Zika virus is currently of great interest for much of the world. The findings are of significant importance due to the fact that these structures are of the whole virus (rather than individual protein components) bound to a neutralizing Fab, which gives it biological relevance. In fact, conclusions like those drawn by the authors in this study would not be possible without studying the whole virus.

The strongest finding in this manuscript by far is that the E protein layer is stabilized by C10 at pH 5. This alone is, to my knowledge, a novel and very interesting finding. The structures provided were of high quality and at high enough resolution (at least for the pH 6.5 and pH 8 structures) to derive meaningful information about potential binding epitopes for the Fab. However, the “controls” presented (unbound ZIKV) were only in the form of micrographs and 2D classes. I think this paper would be far more compelling and more informative if it were to also solve the structures that come from the unbound viruses at low pH. Having actual control

structures like this would be just as novel a finding as the antibody-bound low-pH structures and would provide an additional layer of information that is missing here (i.e., what happens to the unbound ZIKV in 3D during the infection process).

There are several issues, including areas where additional information should be provided, that should be addressed before this paper is suitable for publication.

1. Lines 78-79: “This suggests the E proteins are likely “flopping” on the virus surface.”

The authors conclude that the absence of a resolved E protein layer in Fig. 1 is evidence that the protein layer has become loose and is flopping around, but they do not address the possibility that the protein layer is no longer attached. A gel of the virus sample after incubation at pH 5 would ensure that the E protein layer is still associated with the virus and would support the authors’ claim.

2. Lines 130-131: “These interactions result in the E proteins on virus surface being locked together and are critical for its neutralization mechanism.”

This statement is far too strong. No mutational studies are described to conclusively show which, if any of the estimated epitopes are necessary for neutralization. This is simply the authors’ best guess for how the Fab may neutralize. While the story seems likely, the statement cannot be made so conclusively without actual experimental evidence.

3. Lines 66-68: “We solved the cryoEM structures of Fab C10 complexed with ZIKV at pH 8, pH 6.5 and pH 5...and compared them to the uncomplexed ZIKV controls at respective pH.”

There were no unbound ZIKV structures presented in this paper. By “uncomplexed controls at respective pH”, the authors seem to mean micrographs and 2D classes of uncomplexed virus, but this is not the same as comparing two sets of 3D structures. The wording should be more clear here.

4. Line 76: In the authors’ view, what might the “hair-like densities protruding from the virus surface” be? Are they expected to be E proteins extending from the virus? Please note where these are found in Fig. 1 as well.

5. Line 104: The authors estimated C10-binding epitopes based on a cutoff of 5 Å between side chains of the Fab and E proteins or 8 Å between the C chains of the Fab and E proteins.

Where do the 5 Å and 8 Å values come from? Please indicate how these numbers were chosen. Can you add a visual explanation (perhaps in Fig. 3) as to how these residues were chosen?

6. Lines 64-65: The authors suggest that C10 is likely to neutralize at a post-attachment step based on its ability to prevent ADE of ZIKV infection. Is there any evidence that the Fab can remain attached to ZIKV after endocytosis into a cell? This would validate both this statement and the general premise of the paper, which suggests that the Fab should be able to stabilize E

protein rafts within the endocytic environment of the cell.

7. Why are imaging/processing conditions so different between the pH 5 and the pH 6.5/8 samples? Please explain why the different strategies were used.

8. The authors indicate that they used the EMD-8139 structure as a starting model during processing. Was this structure first filtered to lower resolution to prevent model bias? If not, does it change the resulting structure when a low-resolution model is used?

9. In Figure 1, can the authors please explain the following:

a) In the unbound ZIKV 2D class averages, why are the membrane bilayer and E protein layer much more well-defined at pH 6.5 than at pH 8?

b) In the unbound ZIKV pH 5 2D class average, why might the outer membrane layer become so much wider than at higher pH?

c) In the C10-ZIKV pH 5 2D class average, why do the distance values not reflect the small outward radial shift by the E protein layer (i.e., 240 Å for the E protein layer)?

d) Please indicate with arrows or other markers where the “hair-like densities” described are seen in the micrographs or 2D class average for the pH 5 ZIKV control.

e) Please add a gel to confirm that the E protein layer is still present in the pH 5 unbound ZIKV sample.

10. Why do distance values vary between Fig. 1 (2D class averages), Fig. 4 (panel A) and Extended Data Fig. 1, even when the same structure is compared? In Fig. 1, the membrane bilayer appears at 180-200 Å in all structures, while in Fig. 4, it shows up at 148-199 Å and in Extended Data Fig. 1, it is 160-200 Å. The E protein layer is often shown at the same radius as well, even when the authors claim that it has shifted outward radially.

Reviewer #1 (Remarks to the Author):

This manuscript describes cryo-EM analysis of Zika virus in complex with a neutralizing human monoclonal antibody, that was previously shown to neutralize dengue virus, at different pHs to gain insight into the mechanism of how this cross-reactive antibody neutralizes Zika virus infection in the context of membrane fusion. Authors first provide an analysis of cryo-EM images of the un-complexed native Zika virions at different pHs and show that lowering of pH from 8 to 6.5 radially extends the outer E protein layer, and at pH 5.5, this layer loosens further and extends out from the virion lipid layer. These observations are then compared with cryo-EM reconstructions of Zika virus in complex with the antibody at pH of 8, 6.5, and 5.5. These reconstructions are performed to a resolution of 4.0, 4.4 and 12 Å at these pH values, respectively. They clearly show that the antibody binds to the E protein layer at all these pHs. Comparative analysis of the complex structures at these pHs, and with that of cryo-EM image analysis of uncomplexed virions shows that this particular antibody constrains the movement of the E protein at pH 6.5 and further restricts structural rearrangement of the E protein subunits critical for the membrane fusion event.

The manuscript describes a set of novel and interesting structural results, and raises the possibility of using this kind of a cross-reactive antibody in designing neutralizing immunotherapeutic antibodies that are also effective in preventing the antibody-mediated enhancement (ADE effect). The manuscript is fairly well-written with adequate

methodological details and appropriate discussion. The cryo-EM analysis is well done and results are convincing, and interpretations are reasonable.

My other major comments:

1. There are no structural details of the bound Fab presented in the main text. Authors MUST provide a summary of paratope-epitope interactions in the main text (which CDRs, residues etc) , and a figure showing how well the Fab density is resolved (either in the text or in the extended data). Extended data do show the paratope-epitope interactions, but a figure of close up view of the interacting regions would be useful. It is not clear why closer interactions are not listed, one would think at near ~ 4 Å resolution, with a reasonable atomic model, one would be able to ‘see’ hydrogen bond and other closer interactions.

The CDR loops on the antibody involved in interaction with E proteins is now indicated in Supplementary Table 1. We have shown the fitting of the Fab into the cryoEM density map in Supplementary Figure 5a. The close-up view of the part of the Fab-E protein interface has been provided in Supplementary Figure 5d, showing polar interactions (hydrogen bonds and electrostatic interactions).

2. Provide further cryo-EM imaging details – a) range of defocus values used during imaging and more details about the CTF corrections (important because the

reconstructions at different resolution are compared)

We have added into the method section.

“The images were taken at underfocus in 0.5~2.5 μ m range. The astigmatic defocus parameters were estimated with Gctf²⁵ and accounted for in orientation search and 3D reconstruction procedures in MPSA²⁶ and Relion²⁷.”

and b) how the magnification was calibrated (important because the observations pertain to radial expansions).

The microscope magnification was calibrated using cross-grated grid according to the FEI manufacturer’s procedure. To derive radial expansion of the E protein layer, we make sure that the lipid bilayer membrane of the pH8 and pH 6.5 complex unexpanded and the pH5 complex expanded structure are at the same radius and then compare the positions of their E protein layer. We cannot claim that the lipid membrane between these two structures are exactly the same, but we are sure that the E protein layer of the pH5 complex structure did move away from the lipid membrane layer.

3. Line 149: be more specific – which “another stage”

Changed to “possibly the “open trimeric E protein conformation”¹⁹”.

4. Line: 161: be more specific about which “repelling force at this interface”

Changed to “electrostatic charge repelling force”

5. Include a succinct more cogent discussion at the end that summarizes how the structural observations relate to fusion event sequence mentioned in the introduction.

We changed

“Comparing the pH8 and the pH5 complex structures (Fig. 4b), shows that the maximum radial movement of the E protein outwards is at one end of the A-C’ dimer near the 5-fold vertex ($\sim 15\text{\AA}$). This suggests the membrane associated stem regions of the E protein need not be fully extended (up to $\sim 65\text{\AA}$ in length) for this movement. In sharp contrast, a previous study¹ describing a very low resolution cryoEM map of a DIII-binding Fab E16:WNV complex at pH6, showed the E protein layer moved radially outwards by $\sim 60\text{\AA}$, even though the E protein density was not interpretable. Another low resolution cryoEM structure of antibody E104 complexed with DENV was shown to inhibit another stage of the fusion process².”

To

“Comparing the pH8 and the pH5 complex structures (Fig. 4b), shows that the maximum radial movement of the E protein outwards is at one end of the A-C’ dimer near the 5-fold vertex ($\sim 15\text{\AA}$). This suggests the membrane associated stem regions of the E protein need not be fully extended (up to $\sim 65\text{\AA}$ in length) for this movement. In sharp contrast, a previous study¹ describing a very low resolution cryoEM map of a DIII-binding Fab E16:WNV complex at pH6, showed the E protein layer moved radially outwards by $\sim 60\text{\AA}$, even though the E protein density was not interpretable. **Our pH5 complex structure here shows a smaller radial expansion of the E protein layer and**

therefore may be an even earlier event of fusion process involving the dissociation of the E protein layer from the lipid membrane. Another low resolution cryoEM structure of antibody E104 complexed with DENV was shown to inhibit another stage of the fusion process, **possibly the “open trimeric E protein conformation”². This is likely a step prior to the formation of the closed trimeric E protein structure.”**

6. Consider including comparative radial plots (with both uncomplexed and complexed structures) which would be more effective in illustrating the radial changes.

We have added a figure (Supplementary Figure 1b) showing the radial change of both uncomplexed and complexed structure at different pHs.

Reviewer #3 (Remarks to the Author):

The manuscript titled, “Neutralization mechanism of a highly potent antibody against Zika virus” by Shee-Mei Lok and colleagues describes the cryo-EM structures of the C10-ZIKV complex at different pH levels. While this study is not the first antibody-bound ZIKV structure solved, it represents a novel attempt to describe the antibody-bound structure in a biologically relevant way using varied pH levels to mimic the environment likely to be encountered by the virus during an infection cycle. The authors found that unbound ZIKV goes through structural changes as the pH decreases, showing deformed and aggregated viruses and losing structural stability of the E surface protein layer. However, when the C10 Fab is pre-bound to ZIKV, the E protein layer becomes

stable, even at low pH. The authors estimated Fab-binding epitopes, and hypothesize that the locations are appropriate for stabilizing inter-dimer, intra-dimer and inter-raft protein interfaces. The authors point to this stabilization, as a likely mechanism for virus neutralization by this Fab.

This study is novel and timely, as the Zika virus is currently of great interest for much of the world. The findings are of significant importance due to the fact that these structures are of the whole virus (rather than individual protein components) bound to a neutralizing Fab, which gives it biological relevance. In fact, conclusions like those drawn by the authors in this study would not be possible without studying the whole virus.

The strongest finding in this manuscript by far is that the E protein layer is stabilized by C10 at pH 5. This alone is, to my knowledge, a novel and very interesting finding. The structures provided were of high quality and at high enough resolution (at least for the pH 6.5 and pH 8 structures) to derive meaningful information about potential binding epitopes for the Fab. However, the “controls” presented (unbound ZIKV) were only in the form of micrographs and 2D classes. I think this paper would be far more compelling and more informative if it were to also solve the structures that come from the unbound viruses at low pH. Having actual control structures like this would be just as novel a finding as the antibody-bound low-pH structures and would provide an additional layer of information that is missing here (i.e., what happens to the unbound ZIKV in 3D during the infection process).

It is impossible to solve the structure by cryo-EM because the particles are heterogenous. Attempts to determine orientations are not successful because the class averages always shows a missing E protein layer at pH 5, we do not think that the E protein layer has fallen out of the virus surface as we observed hair-like protrusions which are likely the E proteins. The reason for the missing E protein layer after average is due to the E proteins flopping on the surface.

There are several issues, including areas where additional information should be provided, that should be addressed before this paper is suitable for publication.

1. Lines 78-79: “This suggests the E proteins are likely “flopping” on the virus surface.”

The authors conclude that the absence of a resolved E protein layer in Fig. 1 is evidence that the protein layer has become loose and is flopping around, but they do not address the possibility that the protein layer is no longer attached.

We now provide a closed up cryoEM image of the pH5 particles showing the hair-like protrusions. The virus surface is spiky looking indicating that the E protein did not fall off the virus.

We have changed Figure 1 legend to:

Figure 1 | CryoEM micrographs of the uncomplexed ZIKV control and the Fab C10-ZIKV complex samples at various pH levels. The deformed particles and aggregates are indicated with red and black arrows, respectively. The right upper corner

inset shows a quarter of a 2D class average of the round particles. The E protein layer is indicated with a green arrow, the outer and inner leaflets of the bilayer lipid membrane with cyan arrows. In the pH5 uncomplexed ZIKV control, the E protein layer is missing in the 2D class average. **Bottom right inset in the pH5 uncomplexed ZIKV control, is a median filtered (5x5 pixel) image that showed particles with hair-like protrusions (blue arrow), which are likely the E proteins flopping on the virus surface.** Scale bar is 500Å.

A gel of the virus sample after incubation at pH 5 would ensure that the E protein layer is still associated with the virus and would support the authors' claim.

Even if the E protein has fallen off the virus surface, it will still be in the sample, therefore running a gel would likely not answer the reviewer's question about the existence of E protein still anchoring on the surface. However, we appreciate the reviewer's thought on this.

2. Lines 130-131: "These interactions result in the E proteins on virus surface being locked together and are critical for its neutralization mechanism."

This statement is far too strong. No mutational studies are described to conclusively show which, if any of the estimated epitopes are necessary for neutralization. This is simply the authors' best guess for how the Fab may neutralize. While the story seems likely, the

statement cannot be made so conclusively without actual experimental evidence.

Changed to

“These interactions result in the E proteins on virus surface being locked together and **could be** critical for its neutralization mechanism.”

3. Lines 66-68: “We solved the cryoEM structures of Fab C10 complexed with ZIKV at pH 8, pH 6.5 and pH 5...and compared them to the uncomplexed ZIKV controls at respective pH.”

There were no unbound ZIKV structures presented in this paper. By “uncomplexed controls at respective pH”, the authors seem to mean micrographs and 2D classes of uncomplexed virus, but this is not the same as comparing two sets of 3D structures. The wording should be more clear here.

Changed to

“We solved the cryoEM structures of Fab C10 complexed with ZIKV at pH8, pH6.5 and pH5 mimicking the extra-cellular, early and late endosomal conditions, respectively, and compared them to the cryoEM maps of the uncomplexed ZIKV controls at pH 8³, pH6.5 (Supplementary Fig. 1b) and the 2D-class average of pH5 particles (Fig. 1).”

4. Line 76: In the authors’ view, what might the “hair-like densities protruding from the virus surface” be? Are they expected to be E proteins extending from the virus? Please

note where these are found in Fig. 1 as well.

Yes, there are only E proteins and M proteins on virus surface and M proteins are very small, therefore the hair-like protrusions are most likely E proteins. We now provide a filtered image of the pH5 uncomplexed particles (lower right inset) to show the hair-like protrusions in Fig. 1.

5. Line 104: The authors estimated C10-binding epitopes based on a cutoff of 5 Å between side chains of the Fab and E proteins or 8 Å between the C chains of the Fab and E proteins.

Where do the 5 Å and 8 Å values come from? Please indicate how these numbers were chosen. Can you add a visual explanation (perhaps in Fig. 3) as to how these residues were chosen?

Our cryoEM map is at 4Å resolution, this resolution is a border line case as to whether you can determine hydrogen bonds accurately, because some side chains densities are observed while others are not. Some people do not even accept discussion of hydrogen bond at this resolution, while others are in favor. That is why we identified interactions with both criteria. Of course, the lists of interacting residues are slightly different depending on the criteria used. We selected these values cutoff based on two widely cited papers^{4, 5}, that describe the cutoff when using side-chains interaction (hydrogen bonds/electrostatic interaction: 4Å and hydrophobic interactions: 5 Å), and when using

C-alphas (8 Å). We have added the figure to Supplementary Figure 5d to show hydrogen bonds/electrostatic interactions of part of the interacting interface between the Fab and E protein.

We also make clarifications in the main manuscript and added references:

“In the 4.0Å resolution pH8 complex cryoEM map (Fig. 2a), the likely interacting residues that form the epitope were identified, by using a cutoff of 5Å distance¹⁶ (hydrogen bonds/electrostatic interaction: 4Å and hydrophobic interactions: 5Å) between side chains of the Fab and E proteins (Fig. 3, Supplementary Table 1). We also presented the epitope identified with a cutoff of 8Å distance¹⁷ between the C α chains of the Fab and E proteins (Supplementary Fig. 5c)”

6. Lines 64-65: The authors suggest that C10 is likely to neutralize at a post-attachment step based on its ability to prevent ADE of ZIKV infection. Is there any evidence that the Fab can remain attached to ZIKV after endocytosis into a cell? This would validate both this statement and the general premise of the paper, which suggests that the Fab should be able to stabilize E protein rafts within the endocytic environment of the cell.

Firstly, we solved the complex structure at pH6.5 and 5.0 and they clearly showed Fab densities. These pH conditions mimic the endosomal conditions. One may argue that proteases in the endosome will digest the antibodies, but addition of antibodies did prevent antibody dependent enhancement (Dejnirattisai et al., 2016) suggesting that the

function of the antibody remains.

7. Why are imaging/processing conditions so different between the pH 5 and the pH 6.5/8 samples? Please explain why the different strategies were used.

Usually we can solve structures to about 2-3 times the pixel size (which is determined by the magnification). When we used higher magnification for solving high resolution structure, we also need to collect a lot of data and therefore use a lot of microscope time. Therefore we would only do it, if we think we can get high resolution structure. When the samples look heterogeneous, we know that the chances of getting high resolution structure is very low, we will choose to image at lower magnification so that we will use shorter microscope time and also collect more images so that we can sort particles into different classes.

The pH6.5 and pH8.0 samples appear homogenous and we think high resolution cryoEM structures are achievable and therefore we imaged at high magnification. The pH 5 particles clearly do not look ordered and therefore we choose to image at lower magnification to get more particles, just in case we may need to sort them into different structural classes, which were not detected during reconstruction.

8. The authors indicate that they used the EMD-8139 structure as a starting model during processing. Was this structure first filtered to lower resolution to prevent model bias? If not, does it change the resulting structure when a low-resolution model is used?

For early cycles of orientation searches, we used the signal in reciprocal space only up to resolution of 30Å for both the initial model and the particles for orientation search, this is equivalent to low pass filtering the initial model.

Our initial model is that of the uncomplexed virus, after the first cycle, the Fab densities appeared, we did not put in an initial model for the Fabs and therefore, the Fab densities are not a result of model bias.

9. In Figure 1, can the authors please explain the following:

a) In the unbound ZIKV 2D class averages, why are the membrane bilayer and E protein layer much more well-defined at pH 6.5 than at pH 8?

This is because, at pH8, there are more interactions between the E protein layer and the bilayer lipid membrane (therefore visibly less separation between these two layers). In pH6.5 sample, a slight radial movement of the E protein outwards is detected (Supplementary Fig. 1b). This suggests that the E protein lost interactions with the lipid bilayer membrane, and therefore the increased distance between the E protein layer and the lipid membrane make it more visibly “defined”.

b) In the unbound ZIKV pH 5 2D class average, why might the outer membrane layer become so much wider than at higher pH?

They are likely looser and less homogenous and therefore when averaged they look smeared out.

c) In the C10-ZIKV pH 5 2D class average, why do the distance values not reflect the small outward radial shift by the E protein layer (i.e., 240 Å for the E protein layer)?

We only see small radial difference for C10 complexes in the reconstructed 3D structures (Fig. 4a), and not 2D class averages. This is because in 3D reconstruction, the orientations determined are more accurate due to use of improved model (in the late cycles of orientation searches), in addition we also imposed icosahedral symmetry and that further improves the map. The 2d averages in contrast, are just plain comparison between the particles without the use of accurate models and therefore they are averages of several particles with different orientations. The precision of the radius of particles is therefore less accurate in the 2D averages compared to the 3D maps. The radial expansion is only about 8-15Å and therefore may not be detectable by using 2D class averages.

d) Please indicate with arrows or other markers where the “hair-like densities” described are seen in the micrographs or 2D class average for the pH 5 ZIKV control.

This is now included in Fig. 1, lower right inset in the pH5 ZIKV control micrograph.

e) Please add a gel to confirm that the E protein layer is still present in the pH 5 unbound ZIKV sample.

See answer to reviewer #3, Question(1).

10. Why do distance values vary between Fig. 1 (2D class averages), Fig. 4 (panel A) and Extended Data Fig. 1, even when the same structure is compared? In Fig. 1, the membrane bilayer appears at 180-200 Å in all structures, while in Fig. 4, it shows up at 148-199 Å and in Extended Data Fig. 1, it is 160-200 Å. The E protein layer is often shown at the same radius as well, even when the authors claim that it has shifted outward radially.

For the radial difference between 2D averages (Fig 1) and 3D maps (Fig. 4 and Supplementary Fig. 1b), see answer to reviewer #3, Question 9(c).

For difference between the radius of 3D maps of Fig. 4 and Supplementary Fig. 1a, the cross section of the maps is viewed from a different direction and therefore, their radius are slightly different. To chose this direction to emphasize the difference.

1. Kaufmann B, *et al.* Capturing a flavivirus pre-fusion intermediate. *PLoS Pathog* **5**, e1000672 (2009).
2. Zhang X, *et al.* Structure of acidic pH dengue virus showing the fusogenic glycoprotein trimers. *J Virol* **89**, 743-750 (2015).
3. Kostyuchenko VA, *et al.* Structure of the thermally stable Zika virus. *Nature* **533**, 425-428 (2016).
4. Mukherjee S, Zhang Y. MM-align: a quick algorithm for aligning multiple-chain protein complex structures using iterative dynamic programming. *Nucleic acids research* **37**, e83 (2009).

5. Chung-Jung Tsai SLL, Haim J. Wolfson, Ruth Nussinov. A dataset of protein-protein interfaces generated with a sequence-order-independent comparison technique. *J Mol Biol* **260**, 604-620 (1996).

Reviewers' Comments:

Reviewer #1 (Remarks to the Author):

The revised manuscript adequately addresses all my concerns and the revisions made in response to my comments are appropriate.

Reviewer #3 (Remarks to the Author):

The authors have addressed reviewers' concerns appropriately in this revised manuscript of "Neutralization mechanism of a highly potent antibody against Zika virus". No further concerns need to be addressed, and the manuscript is now suitable for publication.